physiology/image processing/biomechanics

trabecular segmentation, automatic segmentation, cortical bone, trabecular bone, thresholding, Avizo

**Author for correspondence:**
Eva C. Herbst
e-mail: eva.herbst@pim.uzh.ch

# A new straightforward method for semi-automated segmentation of trabecular bone from cortical bone in diverse and challenging morphologies

Eva C. Herbst[1], Alessandro A. Felder[2],
Lucinda A. E. Evans[3], Sara Ajami[4], Behzad Javaheri[5]
and Andrew A. Pitsillides[3]

[1]Palaeontological Institute and Museum, University of Zurich, Zurich, Switzerland
[2]Research Software Development Group, Research IT Services, University College London, London, UK
[3]Skeletal Biology Group, Comparative Biomedical Sciences, The Royal Veterinary College, London, UK
[4]Great Ormond Street Institute of Child Health, University College London, London, UK
[5]School of Mathematics, Computer Science and Engineering, City University of London, London, UK

ECH, 0000-0003-3640-9695; AAF, 0000-0003-3510-9906;
SA, 0000-0003-3313-1632; AAP, 0000-0002-3861-998X

Many physiological, biomechanical, evolutionary and clinical studies that explore skeletal structure and function require successful separation of trabecular from cortical compartments of a bone that has been imaged by X-ray micro-computed tomography (micro-CT) prior to analysis. Separation often involves manual subdivision of these two similarly radio-opaque compartments, which can be time-consuming and subjective. We have developed an objective, semi-automated protocol which reduces user bias and enables straightforward, user-friendly segmentation of trabecular from the cortical bone without requiring sophisticated programming expertise. This method can conveniently be used as a 'recipe' in commercial programmes (Avizo herein) and applied to a variety of datasets. Here, we characterize and share this recipe, and demonstrate its application to a range of murine and human bone types, including normal and osteoarthritic specimens, and bones with distinct embryonic origins and spanning a range of ages. We validate the method by testing inter-user bias during

the scan preparation steps and confirm utility in the architecturally challenging analysis of growing murine epiphyses. We also report details of the recipe, so that other groups can readily re-create a similar method in open access programmes. Our aim is that this method will be adopted widely to create a reproducible and time-efficient method of segmenting trabecular and cortical bone.

## 1. Introduction

Biomechanists, bone physiologists, biologists, clinicians and palaeontologists analyse bone structure to answer a myriad of questions [1–4]. Some whole bone analyses rely on the presumed conservation of development, remodelling and repair factors across cortical and trabecular bone compartments, yet most studies investigate these two bone types separately due to the likelihood that inherent differences in their behaviour and responsiveness exist [5–7]. Compared to the cortex, trabecular bone can constitute a small fraction of total volume, is more porous on the tissue level, and via its large surface area for remodelling supplies most of the exchangeable calcium pool [8]. This illustrates the differing functions and performance of these structurally diverse compartments, which have been reported to extend to bone type-related differences in osteoblast behaviour even when isolated and maintained *in vitro* [9]. Due to these differences, many studies investigate the cortical or trabecular compartments in isolation, which necessitates the effective segmentation of these two types of bone.

For computed tomography (CT) data, a manual segmentation step is commonly used to differentiate cortical and trabecular bone [10–14], but is both time-intensive and subjective. In particular, it is difficult to determine where the cortical bone ends and the trabecular bone begins. The guidelines for defining trabecular boundaries in manual segmentations are rarely reported [11–13,15]. Another commonly used alternative to manual segmentation is to crop a region of interest (for example, a sphere or cube) from the trabecular region and analyse only this volume [2,16]. However, this approach excludes data from the trabecular regions outside of this volume, making it likely that changes near the cortical boundary would, therefore, be missed.

Previous studies have also developed automated segmentation protocols, but they have various limitations or are time-consuming to implement. Methods avoiding manual segmentation of two-dimensional slices use bimodal methods that rely on single grey value thresholding, which tend to fail in complex scenarios [17] or gradient-based edge detection algorithms which perform well [18,19]. For example, Lublinsky *et al.* [20] built a 5-step freely available algorithm that mapped the periosteal edge to create a cortical mask, the innermost edge of which defined the interface between the cortical and trabecular bone. This, however, requires specific definition of a filter, threshold and categorization of cortical thickness (thick or thin) prior to analysis, and that the volume of the space outside the bone exceeds the volume of the inter-trabecular space, thus restricting its utility to only some CT images. Other methods require constant cortical bone thickness (e.g. [14]) or require that this border is continuous and thicker than that of the trabeculae [21], assumptions that are not valid in many bones.

An alternative approach used a statistical shape model for automatic segmentation, but this approach requires the development of a training set of segmented images and assumes that sufficient reference specimens are available, which is not always the case [4]. Too much variation between samples can also cause issues in the trabecular segmentation ([4, fig. 7]). Another exciting method to automate segmentation for FE modes [22] applied the Stradwin software, which calculates cortical thickness based on the intensity normal to the periosteal surface ([23]; note this software has now been replaced by Stradview). In this automated approach, one specimen was segmented and used as a template, which was then warped onto target samples to determine the periosteal surface, which was used as an input by the Stradwin software to run the cortical bone thickness calculation and determine the endosteal surface. Creation of a template may not, however, always be possible (e.g. in fossils where no other specimens exist). The Stradwin software can also be used without a template to segment bone [23], but this requires manual editing of contours to generate the periosteal surface. This approach also assumes differing attenuation values between the trabecular and cortical bone, which is not always the case (e.g. in the specimens used herein). The methods by Buie *et al.* [24], built upon by Burghardt *et al.* [25], solve many of these issues. These methods are, however, designed for specific morphologies and require certain conditions about the bone distribution to be met, specifically relating to the diameter (in pixels) of the largest pore (or foramen) connecting the marrow to the exterior.

To address the limitations in other segmentation methods, we have developed an objective, semi-automated protocol for segmenting trabecular and cortical bone which was validated in animal and human bone samples and is flexible enough to be applicable across various challenging morphologies. This is achieved by automated trabecular segmentation within a semi-automated overall process in which a simple preprocessing step is required to isolate the region of interest. Our method relies on an input segmentation of the marrow space and can easily be loaded as a 'recipe' in Avizo, which enables all further steps to run automatically. We compare this to other automatic segmentation methods to build upon previous advances.

# 2. Material and methods

## 2.1. Micro-CT datasets

The automated segmentation algorithm was initially evaluated in knee joint epiphyses of normal healthy control (CBA) and osteoarthritis-prone (STR/Ort) mouse strains, followed by samples from the human femoral head, skull and vertebrae. The micro-CT dataset originated from: (i) tibial epiphysis of a skeletally mature 20-week-old normal CBA mouse, (ii) tibial epiphysis of a 19-week-old osteoarthritis-prone STR/Ort mouse, (iii) tibial epiphysis of an ageing 51-week-old osteoarthritis-prone STR/Ort mouse, (iv) part of a human femoral head from a 77-year-old female, (v) human vertebra from a 24-year-old female (data from [26]), (vi) human parietal bone from a five-month-old male with craniosynostosis and (vii) tibial epiphysis of a young, growing eight-week-old STR/Ort mouse. The latter two specimens were chosen specifically for the high porosity and foramina content of their developing primary cortical bone. Scanning parameters and links to datasets are reported in table 1. All images were 8 bit. The human samples were retrieved following ethical approval and patient consent. Ethical approval for mouse samples was carried out in accordance with the Animals (Scientific Procedures) Act 1986, approved by the Royal Veterinary College Ethical Review Committee and the United Kingdom Government Home Office (permit P25338BB2). Our CT scan data are available in the following Figshare repository: https://figshare.com/projects/Trabecular_and_Cortical_Bone_Segmentation_Method/99434.

## 2.2. Application of algorithm to the three-dimensional dataset

Datasets first required some semi-automated preprocessing to segment the marrow space, which was then used as the input for the trabecular segmentation algorithm. This preprocessing step and algorithm implementation was performed in Avizo (Thermo-Scientific, v. 2019.2 and 2020.1).

### 2.2.1. Preprocessing of dataset

We preprocessed the dataset in Avizo, although other segmentation software could also be used. First, we filtered the CT scans using a non-local means filter (three-dimensional GPU adaptive manifold setting) to remove noise in the scans and facilitate the watershed operation (figure 1a). Next, we separated the bone of interest from the rest of the scan by placing 'seeds' and assigning them to two different regions (defined as 'materials' in Avizo): the first containing both the bone of interest and its marrow space, and the second being the background (any bone(s) not included in analysis and background voxels) (figure 1b). Then a watershed operation was performed on these 'seeds' (figure 1c). The watershed algorithm assigns regions based upon the positioning of these different 'seeds', using changes in the gradient of the voxel greyscale values to determine the boundaries of these regions. For the murine samples, the bone of interest was the entire tibial epiphysis (figure 1a), which was removed from the metaphysis by the watershed operation (figure 1c). For the human samples, all bone present in the scan was included in the analysis.

Growth plate bone bridges (radiopaque tissue connecting the metaphysis and the epiphysis, see [27]) were manually removed from the tibial epiphysis. For the human samples, the bone and marrow space were separated from the background using the same approach, although the growth bridge removal was not required. The scans of the human bone samples were, therefore, much quicker to pre-process. This preprocessing was followed by the use of appropriate thresholding ranges chosen manually to separate the marrow space from the bone (table 1). The marrow space is the input for the automatic method. This preprocessing step enabled a semi-automated segmentation of the whole region of interest (mouse epiphysis in figure 1c) that was quicker and less biased than manual segmentation.

**Table 1.** Scanning parameters and links to datasets used in the study. Thresholds are greyscale thresholds for 8 bit images.

| dataset | scanner | voxel size (mm) | kV | mA | marrow space threshold | data availability |
|---|---|---|---|---|---|---|
| STR/Ort mouse tibial epiphysis (8 weeks) | Bruker Skyscan1172 | 0.005 | 49 | 200 | 0–70 | https://doi.org/10.6084/m9.figshare.14141159 |
| STR/Ort mouse tibial epiphysis (19 weeks) | Bruker Skyscan1172 | 0.005 | 49 | 200 | 0–70 | https://doi.org/10.6084/m9.figshare.14097983 |
| STR/Ort mouse tibial epiphysis (51 weeks) | Bruker Skyscan1172 | 0.005 | 49 | 200 | 0–70 | https://doi.org/10.6084/m9.figshare.14098052 |
| CBA mouse tibial epiphysis (20 weeks) | Bruker Skyscan1172 | 0.005 | 49 | 200 | 0–70 | https://doi.org/10.6084/m9.figshare.14097380 |
| human femoral head section (77 years) | Bruker Skyscan1172 | 0.01 | 80 | 124 | 0–37 | https://doi.org/10.6084/m9.figshare.14097263 |
| human parietal section (5 months) | Bruker Skyscan1172 | 0.009 | 49 | 200 | 0–49 | https://doi.org/10.6084/m9.figshare.14135879 |
| human vertebra section | unknown—images from Figshare | 0.03 | 90 | unknown—images from Figshare | 0–30 | Mills & Boyde [26]: https://doi.org/10.6084/m9. figshare.1196897.v1 |

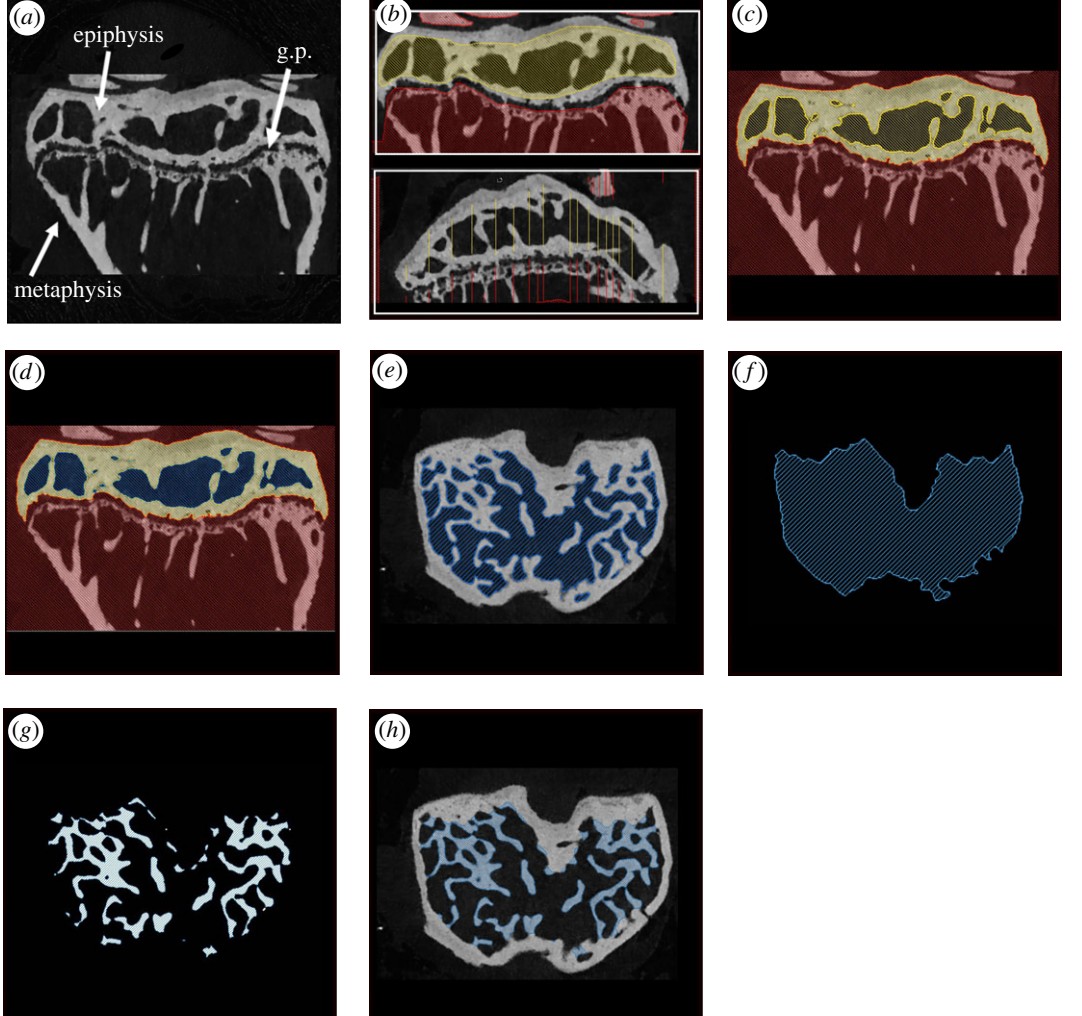

**Figure 1.** Preparation and step-wise application of algorithm, demonstrated within the tibial epiphysis of a 19-week-old STR/Ort mouse. (a) Filtered CT scan of mouse proximal tibia (g.p., growth plate); (b) example of seed placement (preprocessing) in the mediolateral (top) and anteroposterior (bottom) slices; (c) result of the watershed operation, with the epiphysis (yellow) separated from the background (red); (d,e) marrow space (blue) separated from the epiphyseal bone via thresholding of greyscale values; (f) the marrow space is shrink-wrapped via the closing algorithm; (g) the trabeculae are isolated by subtracting the marrow space (e) from the shrink-wrapped space (f); (h) the resulting trabecular segmentation shown on the original scan. In (b), note that it is not necessary for the seeds to go all the way to the boundary of the epiphysis but they must, however, include both some bone and some marrow space.

## 2.2.2. Segmentation of trabecular bone

The algorithm consists of several sequential steps applied to a dataset (figure 1a) after isolating the marrow space (figure 1d, see §2.2.1). The algorithm starts by smoothing the input marrow space segmentation (figure 1e), 'shrink-wrapping' the smoothed marrow space (figure 1f), reducing noise by another smoothing operation, and finally automatically selecting the bone inside the smoothed shrink-wrapped volume (by subtracting the marrow space eroded by 3 pixels from the 'shrink-wrapped' region) (figure 1g). This bone corresponds to the trabecular compartment (figure 1h). The first smoothing step is achieved by a 3 pixel ball erosion and a 3 pixel ball dilation. This step removes the small foramina connecting the marrow space to the exterior. These need to be removed so that they are not included in the subsequent shrink-wrapping step. In the 'shrink-wrapping' (figure 1f) step, the marrow space is essentially 'grown' over concavities so that it includes the trabeculae, using a 'ball-closing' operation with a value of 25. This value was based on iterative tests on the mouse samples and worked well for the human samples (which had different resolutions). The closing operation takes a binary image and fills small holes, smoothes object boundaries and connects close

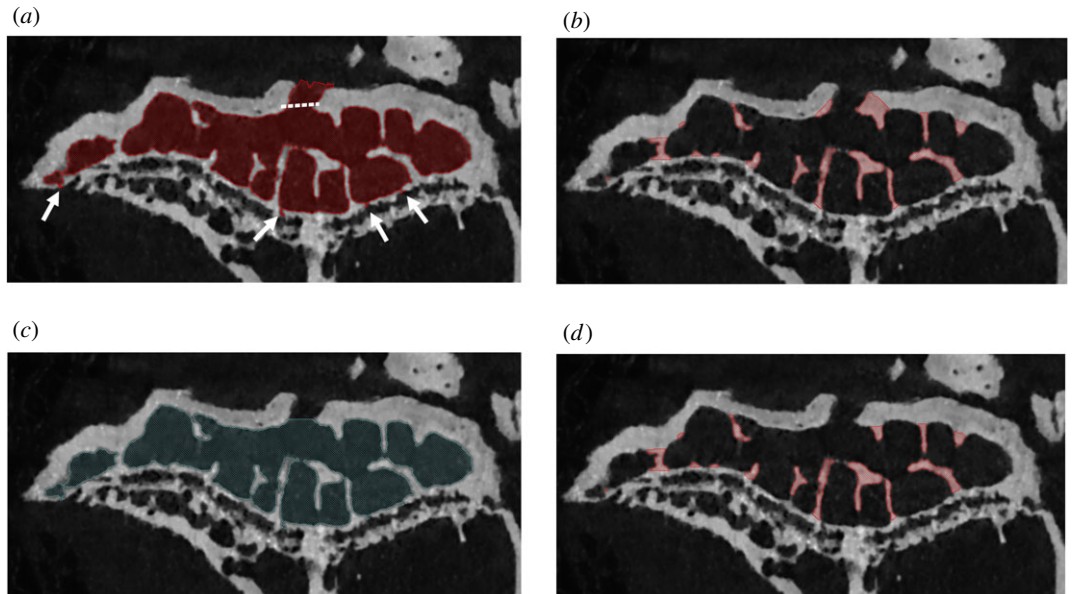

**Figure 2.** Sealing of foramina demonstrated in the tibial epiphysis of a 51-week-old STR/Ort mouse. (*a*) The marrow space without foramen sealing, with the resulting trabeculae shown in (*b*). In (*c*), the marrow space was manually sealed (along dotted line shown in *a*). (*d*) The resultant marrow space following sealing. Very small foramina (see arrows in *a*), are automatically removed by the algorithm to prevent them from being shrink-wrapped. Scale bar, 1 mm.

objects to produce a new binary image. The second smoothing step is achieved by a 1 pixel ball erosion and 1 pixel ball dilation to reduce noise and remove aberrant single pixels. This algorithm was implemented as an Avizo recipe and applied to the micro-CT datasets described above. The code is available here: https://zenodo.org/badge/latestdoi/341621577.

## 2.3. Method for addressing the presence of intra-cortical foramina and porosity

We tested the effect of including foramina that connect marrow space to the epiphyseal exterior before applying the algorithm. These foramina are usually included when the watershed operation segments the bone together with the associated marrow space from the surrounding background (figure 2*a*). When included, the algorithm shrink wraps the foramina and, therefore, inappropriately designates adjacent cortical bone as trabecular bone (figure 2*b*). This can be avoided by manual removal (referred to from now on as 'sealing') of foramina from the inter-trabecular space layer, to the extent of a boundary drawn through the main marrow cavity in the adjacent slices (figure 2*c,d*); small foramina (less than 3 pixel width) are removed by the automatic algorithm (figure 2*a*).

We ran the algorithm on foramina-sealed and non-sealed versions (referred to from now on simply as 'sealed' and 'non-sealed' versions) of 19-week-old and 51-week-old STR/Ort tibial epiphyses. To assess the effect of the sealing step, we compared the relative difference (difference between values obtained from sealed and non-sealed versions as a percentage of sealed version, to nearest 0.01%) for trabecular BV/TV, trabecular volume/cortical volume and anisotropy. We also anticipated that foramina-sealing would have less impact on results in larger bones, since foramina in these bones will be smaller relative to total bone volume. To test this hypothesis, we also analysed part of a human vertebra (24-year-old female, data from [26]) and compared outputs from sealed and non-sealed versions and compared the relative difference (%) for the same parameters. To gauge the reproducibility of our segmentation, we additionally used 'intersection over union' [28] as a metric to quantify overlap between segmentations performed both before and after foramina-sealing. Intersection over union was calculated with a custom ImageJ [29,30] (https://doi.org/10.5281/zenodo.1427262) macro, available as part of https://zenodo.org/badge/latestdoi/341621577.

To determine any potential effects of increased cortical porosity on the functionality of our recipe, we examined the outcome when the recipe was applied to the human parietal bone of a five-month-old male and the tibial epiphysis of an eight-week-old STR/Ort mouse; the cortical bone compartment of these immature bones is particularly porous. Cortical porosity in the human sample consists largely of growing primary cortical bone, with non-consolidated osteons in the very early stages of in-filling. In

**Table 2.** Testing the effects of lower resolution images by applying the algorithm to downsampled versions of the 20-week-old CBA mouse dataset.

| downsampling from original | expected average trabecular thickness (pixels) | algorithm value initial erode dilate | algorithm value closing | algorithm value final erode dilate |
|---|---|---|---|---|
| none (original) | 13.29 | 3 | 25 | 1 |
| ×2 | 6.645 | 1 | 12 | 1 |
| ×4 | 3.3225 | 0 | 6 | 0 |
| ×8 | 1.66125 | 0 | 3 | 0 |

the mouse, the thick porous layer of cortical bone could also be interpreted as a much thinner layer of cortical bone with thick trabeculae; a potential source of high inter-user variation in studies lacking an objective segmentation method. We used these samples to test how our algorithm draws the boundary for such specimens with high porosity. We also used these samples to test some manual adjustments (foramina-sealing), to determine how this affects the boundary definition between trabecular and cortical bone.

## 2.4. Validation: testing inter-user variation

Although our algorithm is automatic, user-input was required in preprocessing CT images. An example is the isolation of epiphyses from entire long bones in mouse samples. However, as noted above with regard to foramen-sealing, the benefit of validating the method on such difficult scans is that we anticipate inter-user bias to be even smaller on scans requiring less preparatory manual preprocessing.

To calculate inter-user bias, two experienced users segmented the same scans; one scan of a 19-week-old STR/Ort mouse tibial epiphysis and one scan of part of a human vertebra (24-year-old female, data from [26]) using the semi-automatic algorithm. In the mouse samples, both users tested the method with and without foramen-sealing. For the human vertebra, both users independently sealed foramina and % difference between users determined. We also used 'intersection over union' [28] as above to quantify the overlap between segmentations performed by different users (E.C.H. and A.A.F.).

## 2.5. Testing recipe on low-resolution scans

We tested the effects of using lower resolution datasets in our 20-week-old CBA mouse epiphysis (table 2) by downsampling (2-, 4- and 8-fold) both the images and the marrow space segmentation, to keep the algorithm input consistent except for the resolution. Downsampling was performed on the marrow space using the 'resample' function in Avizo, which averages a given amount of pixels to produce a lower resolution image. The CT image was then downsampled using the 'resample' module with a box filter using the downsampled marrow space as reference input; this produced the same amount of pixels for both image stacks. The algorithm values were reduced by approximately the same factor that the images were downsampled to account for reduced pixels per feature (exact halving was not always possible because these algorithms do not work with pixel fractions). For lower resolution scans, erosion and dilation steps were set to 0 because even a single pixel in these steps would be a lot relative to the feature size (whereas in the larger scans, these erosion dilation scans were for removal of noise and small foramina).

# 3. Results

Our semi-automatic segmentation method performed well on tibial epiphyses of both healthy ageing control (CBA) and osteoarthritis-prone STR/Ort mouse tibial epiphyses at a range of different ages (figure 3, Supplementary video 1). The algorithm also successfully segmented cortex from trabeculae in parts of a human vertebra, a human femur and a human skull bone (figure 4). The method successfully segmented trabecular bone from cortical bone for scans with a range of different scan resolutions (0.005–0.03 mm). While the preprocessing time varies depending on foramina-sealing and quantity of growth bridges that need to be removed, the computation time of the algorithm was

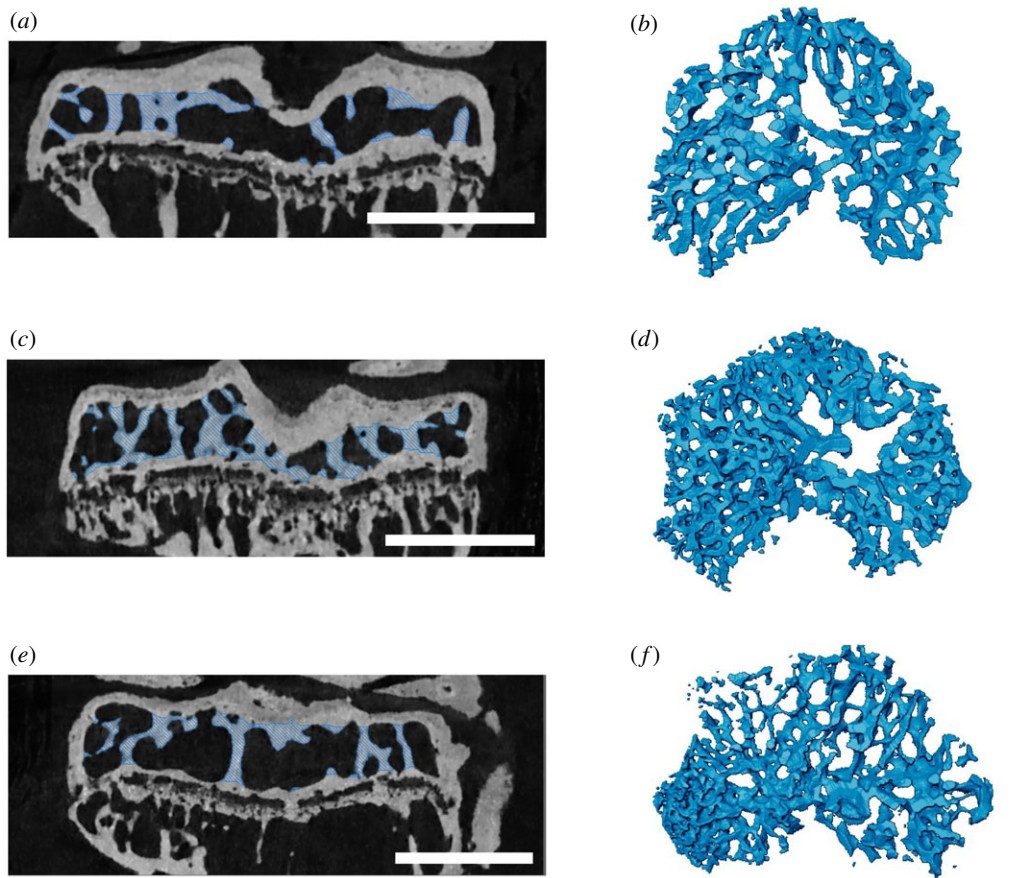

**Figure 3.** Examples of algorithm applied to mouse tibial epiphyses. (*a,b*) Twenty-week-old CBA mouse; (*c,d*) 19-week-old osteoarthritic (STR/Ort) mouse; (*e,f*) 51-week-old osteoarthritic (STR/Ort) mouse. (*a,c,e*) CT scan cross-sections with trabeculae segmented out; (*b,d,f*) three-dimensional volume rendering of segmented trabeculae. Scale bar, 1 mm.

consistently quick (under 30 s with a 32 GB RAM computer with Intel®_Core™_i7-9750H_CPU_at_2.60 GHz and Nvidia GeForce gtx 1650 GPU).

## 3.1. Minor effects of omitting foramina-sealing step

Differences between samples analysed pre/post-foramen-sealing step and 'intersection over union' values are reported in tables 3 and 4, respectively. As predicted, differences between sealed/non-sealed results were minimal, and the two versions differed least in human vertebra where foramina volume was particularly small relative to the overall volume. This sensitivity analysis demonstrates that both for the murine epiphyses and human samples, the foramen-sealing step could be omitted with minimum change to results, which would also be expected for other sample types with similar foramen volume to total volume ratios.

## 3.2. Specimens with high porosity

In human parietal bone, our algorithm first shrink-wrapped a large part of the developing primary cortical bone, classifying solely the most periosteal layer as the cortex. This was due to the osteonal pores being connected and of similar size to trabecular pores. To adjust for this, we manually removed the primary osteonal pores from the marrow space segmentation (figure 5a,b). In the murine epiphysis (figure 5c,d), most foramina were removed by the algorithm and only a few large foramina were manually sealed (see §2.3 and figure 2). We tested whether the inclusion of smaller spaces (such as the foramen marked with a red arrow in figure 5e) affects the trabecular segmentation outcome. There was no effect (0% difference in trabecular volume), because the space was small enough to be automatically removed by the erosion and dilation algorithm steps. Both inputs produced the

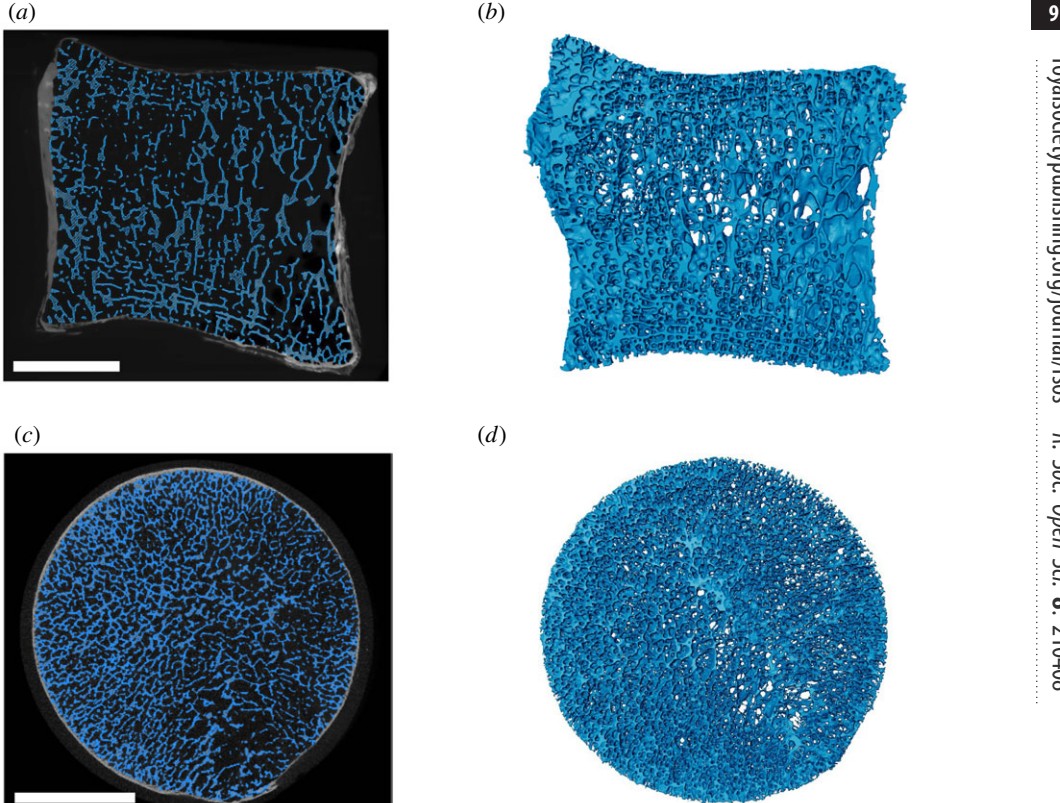

**Figure 4.** Examples of algorithm applied to human samples. (*a,b*) Part of vertebra from a 24-year-old human female [26]; (*c,d*) part of human femoral head from a 77-year-old female. (*a,c*) CT scan cross-sections with segmented trabeculae highlighted in blue and cortical bone left in greyscale; (*b,d*) three-dimensional volume rendering of segmented trabeculae. Scale bar, 10 mm.

**Table 3.** Percentage differences between foramina-sealed/non-sealed samples, and percentage difference between different users (to nearest 0.01%).

| specimen | trabecular BV/TV | trabecular volume/cortical volume | degree of anisotropy |
|---|---|---|---|
| human vertebra (24 years) | | | |
| foramina-sealing effect 1 user | 0.02% | 0.09% | −0.01% |
| inter-user bias foramina-sealing | −0.14% | −12.93% | 0.14% |
| STR/Ort mouse epiphysis (19 weeks) | | | |
| foramina-sealing effect 1 user | 0.20% | 0.58% | −0.03% |
| inter-user bias foramina-sealing | −1.09% | 2.41% | −0.26% |
| inter-user bias no foramen-sealing | −1.32% | 0.32% | −0.30% |
| STR/Ort mouse epiphysis (51 weeks) | | | |
| foramina-sealing effect 1 user | 0.10% | 0.19% | 0.03% |

trabeculae shown in figure 5*f*, confirming that the trabecular boundary segmentation is insensitive to small cortical pores.

## 3.3. Low inter-user bias

Inter-user differences were generally minor between users (less than 2.5%) (tables 3 and 4) except for the trabecular volume/cortical volume in human bone, which was 12.9% due to differences in cortical segmentation, since the trabecular BV/TV anisotropy was very similar between users (less than 0.2%). Closer interrogation of data revealed that the difference in cortical bone was due to one user including

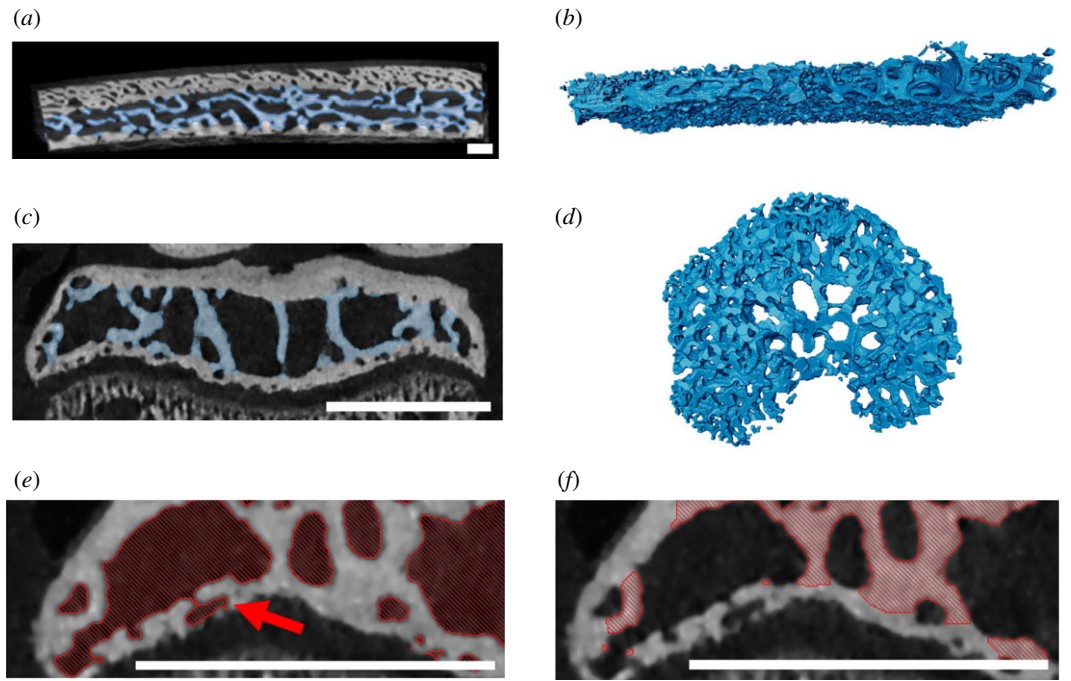

**Figure 5.** Examples of specimens with high porosity. (*a,b*) Section of human skull bone from a five-month-old male with craniosynostosis; (*c–f*) eight-week-old STR/Ort mouse. (*a,c*) CT scan cross-sections with trabeculae segmented out; (*b,d*) three-dimensional model of segmented trabeculae. In the human skull, foramina in the cortical bone were manually removed to ensure that none of the cortical bone was shrink-wrapped. In the mouse, including or manually removing the small horizontally oriented space (red arrow) as part of the marrow space (*e*) did not affect the results; both versions produced the trabeculae shown in (*f*). This is because this space was not shrink-wrapped with the parameters of our algorithm; it was removed during the erosion step. (*f*) Close-up of the trabecular segmentation created after running the algorithm on (*e*). Scale bar, 1 mm.

**Table 4.** 'Intersection over union' values show correspondence of greater than 90% between users, and approximately 99% between foramina-sealed/non-sealed examples for trabecular segmentations (rounded to nearest 0.01%).

| specimen | intersection over union |
| --- | --- |
| human vertebra (24 years) | |
| foramina-sealing effect 1 user | 99.04% |
| inter-user bias foramina-sealing | 99.96% |
| | |
| STR/Ort mouse epiphysis (19 weeks) | |
| foramina-sealing effect 1 user | 91.60% |
| inter-user bias foramina-sealing | 92.03% |
| inter-user bias no foramen-sealing | 99.53% |
| | |
| STR/Ort mouse epiphysis (51 weeks) | |
| foramina-sealing effect 1 user | 99.84% |

a calcified ligament and another user not including this ligament. 'Intersection over union' values show high inter-user correspondence (table 4).

## 3.4. Testing recipe on low-resolution scans

Downsampling in the 20 week CBA mouse dataset slightly impaired algorithm output but generally the trabeculae looked reasonable. In the 2× downsampled version, some small trabeculae near the cortex were excluded from appropriate segmentation (figure 6*c,d*), while in the 4× and 8× downsampled versions

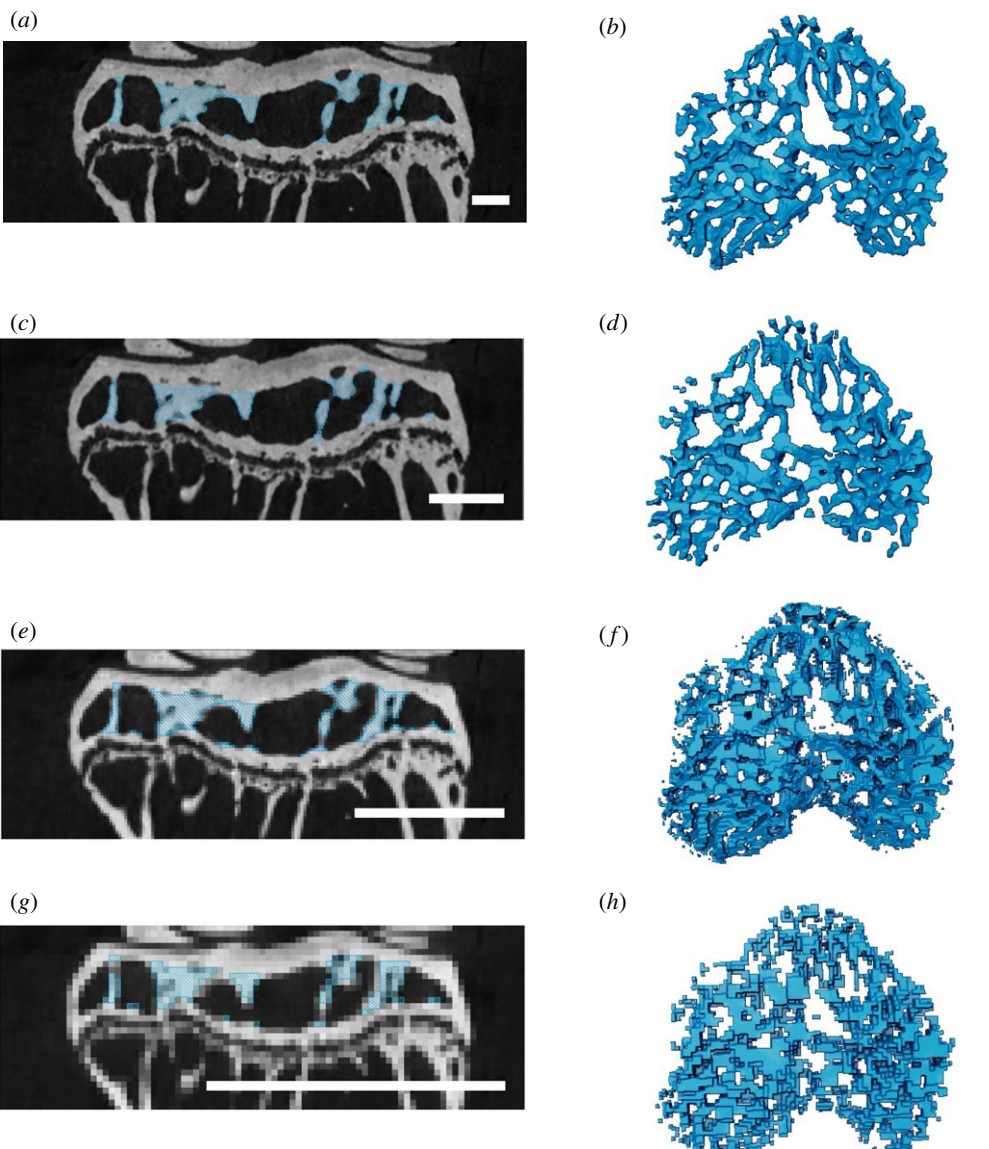

**Figure 6.** The effects of lower resolution images by applying the algorithm to downsampled versions of the 20-week-old CBA mouse dataset. (*a,b*) Original high-resolution image with average trabecular thickness of approximately 13 pixels; (*c,d*) 2× downsampled dataset with the average trabecular thickness of approximately 6.6 pixels; (*e,f*) 4× downsampled dataset with the average trabecular thickness of approximately 3.3 pixels; (*g,h*) 8× downsampled dataset with the average trabecular thickness of approximately 1.7 pixels. More parameters are given in table 2. (*a,d,e,g*) Mediolateral cross-section in the middle of scan. (*b,f,f,h*) Volume rendering of segmented trabeculae. Scale bar, 50 pixels.

(figure 6*e*–*h*), the trabeculae were actually oversampled/noisier relative to the original high-resolution version (figure 6*a,b*). This is because we adjusted the recipe by approximately the same factor as the downsampling, but since subpixel erosions and dilations were not possible, we changed the erode and dilate steps to 0 in the 4× and 8× downsampled images. Including a 1 pixel ball erosion and dilation would have removed too many of the trabeculae in these smaller resolution scans, but works well for removing noise in the higher resolution scans. At the 8× downsampled version, where we expect the average trabecular thickness is only about 1.66 pixels (based on the high res reference image, table 2), the algorithm still produced a rough trabecular segmentation, although the segmentation was not very precise (figure 6*g*).

## 4. Discussion

Our results demonstrate that this method can be applied to bones from a variety of species, of both endochondral and intramembranous origins, and across a range of healthy and pathological

specimens. We anticipate that the application of our new method to further samples from these anatomical compartments and to an even greater range of bones will endorse its utility further. Although this method is not the 'only' semi-automatic segmentation method, it is nonetheless of value to our field because it requires no coding background in users, and is, therefore, accessible and easy to use. Avizo is one of the most commonly used segmentation programmes by researchers and creating an Avizo 'recipe' for this method makes it very easy to implement for Avizo users in particular. Loading our recipe in Avizo does not require any coding knowledge; after downloading the 'recipe', the user selects the preprocessed marrow space as the recipe input and all steps will be run in sequence. In addition to making the Avizo 'recipe' freely available, we share its specific steps in a Github repository (https://github.com/evaherbst/Trabecular_Segmentation_Avizo) so that the method can also be applied in open-source programmes which are more accessible and may be preferred by other researchers.

Usefully, our algorithm works on specimens that would not satisfy the assumptions required for the application of other automated segmentation methods. For example, Ang et al. [21] developed a fast segmentation method (relative to manual segmentation) using thickness ranges to differentiate cortical and trabecular bone. This method would likely overestimate cortical bone in our mouse epiphysis and human vertebra samples, since it would interpret the thick trabeculae adjacent to the cortical bone as actually being cortical bone. Furthermore, gaps and small convex regions are also present in our specimens and could cause underestimation of the cortex in other regions. These types of errors are discussed by Ang et al. [21]; however, future studies are needed to test the extent to which this would be an issue in our samples. The Ang et al. [21] algorithm also differs from ours in that it determines the cortical bone using a thickness map where a circle is fit to the region; their smoothing step removes most of the circular irregularities produced by this, but our algorithm avoids such circular irregularities altogether. Our algorithm is also independent of extra-epiphyseal volume—unlike the automatic algorithm by Lublinsky et al. [20], which requires that the volume outside exceeds the marrow space volume.

It is relevant to note that the Buie et al. [24] method is conceptually very similar to ours. Burghardt et al. [25] built on these methods, providing a programme for implementation. We nonetheless consider both methods valuable and our independent development of a conceptually similar approach highlights the usefulness of all of these methods. Furthermore, our method may reach a different user base, specifically users of Avizo software.

Our method should not cause clipping issues of the outer cortical boundary. Prior methods [24,25] were limited somewhat by the fact that if the space outside of the region of interest is smaller than the dilation and erosion amounts, clipping can occur when the dilation performed to create a 'cortical mask' grows beyond the boundaries of the image. Our method alleviates this limitation by performing the erosion and dilation on the marrow space only and not the cortical bone, leaving the periosteal boundary of the segmented cortex intact.

Our 'closing' step is similar to the Buie et al. [24] erosion/dilation steps, although we first remove small foramina by a separate three-point erosion/dilation of the marrow space, and then apply the closing operation (essentially a larger dilation/erosion step). Our methods mainly differ, however, in not applying dilation and erosion steps on both the cortex and the marrow space to delineate the periosteal and endosteal cortical surfaces, which are then used to create a 'mask' to segment the trabeculae. While these steps in the Buie et al. [24] method typically enable a more automatic detection of marrow space, they also render the method less flexible. Buie et al. [24] noted the issues with large Volkmann's canals and discussed that these need to be 'closed' for the algorithm to work well; this can be solved with a higher amount of erosion and dilation, but this can result in too much smoothing, obscuring small features. The authors proposed a solution in which the periosteal threshold is decreased to create a larger mask; however, these steps require an iterative recipe adjustment: testing to see whether erosion and dilation values used effectively seal the foramina, and then going back to adjust the thresholds to prevent any over-smoothing engendered by large dilations and erosions. Our method (validated on especially difficult mouse epiphyseal samples, where the foramina are large relative to the total volume) enables manual adjustment of the marrow space (e.g. by sealing foramina). This advantage allowed us to deal appropriately with the more complex situations encountered in the human skull and young mice, where a good marrow space segmentation to input into the algorithm is not straightforward to obtain. While Burghardt et al. [25] also defined manual adjustments which could be applied to bones containing either dense trabecular networks or endosteal trabecularization, our method describes in detail how to conduct the foramina-sealing, and tested the effects of including or excluding these foramina.

Our sensitivity analyses showed that differences between sealed/non-sealed images were very low; this manual step can be omitted with only minute change (less than 0.6%) to trabecular BV/TV, trabecular volume/cortical volume and trabecular degree of anisotropy. The degree of anisotropy was the parameter least affected by foramina-sealing (less than 0.04% for all samples) and inter-user differences were generally very low, with only variation in trabecular volume/cortical for the partial human vertebra differing significantly. Closer scrutiny revealed that this was entirely attributable to a calcified ligament attachment site on the outer cortex which was included as the cortex by only one user. This demonstrates that bias was introduced by variations in anatomical knowledge, and not the algorithm's cortical/trabecular separation. This highlights that similar problems could also arise anywhere that densities (greyscale value) of mineralized soft tissues resemble those in bone, particularly entheses. It is, therefore, recommended that users ensure that the objectives of any given study and anatomy of scanned tissues are understood by all participants to ensure accurate 'seed' placement during the watershed step. Our method also enables adjustment of initial erosions and dilations to control the size of automatically excluded foramina, which can be useful for specimens with highly porous cortices. It is imperative that any such deviations from the standard 'recipe' are, therefore, highlighted in any publications applying our algorithm or variations thereof, to ensure that adjustments for different samples are appropriately shared.

## 4.1. Considerations for application

We recommend three visual checks: initially, after the watershed that separates the region of interest, then another to establish that marrow space is isolated from large intra-cortical foramina before running the algorithm (if sinus cleaning is desired), and a third to ensure that the method successfully distinguishes trabecular and cortical bone well for the taxon, element of interest and scan resolution. Users need also to ensure voxel size is not too large relative to structure size, e.g. in our study of murine bone, the voxel size was 0.005 mm and average trabecular thickness was approximately 0.065 mm. Our exact recipe as written will not work on scans with a low feature size relative to the voxel size, as important morphology would effectively be erased during the erosion/dilation (noise removal) algorithm step. However, flexibility in our method will enable users to easily adjust the values used during these steps to fine tune the recipe for their dataset. Such adjustment was indeed employed in our downsampling tests, where scaling of the algorithm inputs by a similar factor to the image downsampling was able to produce reasonable looking segmentations for trabeculae at lower resolution scans without any trial and error tests of the recipe parameters. Further adjustments (for example, testing different erosion and dilation tools, e.g. 'cube' versus 'ball' dilations in Avizo) could improve segmentation of low-resolution scans. Similar downsampling tests could also be used in the future to yield more quantitative comparison with earlier methods [24].

Finally, our original tests on trabecular bone included a few floating pixels of space, as a rare occurrence, near the shrink-wrapping boundary when, for example, a large foramen was not sealed. This was readily remedied by thresholding the trabecular layer to include only bone, using the same bone—space greyscale threshold initially used to isolate the marrow space. Regarding the initial determination of the thresholding values to isolate these marrow spaces, future studies could integrate Otsu thresholding ('factorization' in Avizo's auto-thresholding tool) into our method to automatically threshold marrow space. While this could simplify the method further, as long as the same threshold is used on all datasets in a study, any appropriate threshold (whether chosen manually or automatically) will work.

# 5. Conclusion

Our algorithm is a quick, repeatable method of segmenting trabecular and cortical bone. It can easily be implemented in Avizo, and can also be adopted in open-source programs such as ImageJ. Our algorithm was rigorously tested on a variety of specimens and outputs are robust towards inter-user bias. It does not need any initial segmented reference images, making it ideal for fossil work or for starting on a new project where no training set for the specimens of interest exists. The flexibility of our manual pre-processing steps enables the method to work on a range of morphologies and accommodates targeted adjustments. These could be excluding mineralized ligaments connected to the cortex (seen here e.g. in the human vertebra), excluding cortical primary osteonal bone from being classified as trabecular, and including trabecular bone not enclosed by a cortical shell in the trabecular segmentation (both

seen here e.g. in the human skull fragment). We hope that users can replicate the methods we have specified herein in other, specifically open access software.

Ethics. The human samples were retrieved following ethical approval and patient consent. Ethical approval for mouse samples was carried out in accordance with the Animals (Scientific Procedures) Act 1986, an Act of Parliament of the United Kingdom, approved by the Royal Veterinary College Ethical Review Committee and the United Kingdom Government Home Office (permit P25338BB2).

Data accessibility. The Avizo recipe and instructions for implementation are available on our Github repository: https://github.com/evaherbst/Trabecular_Segmentation_Avizo. If you use this recipe, cite this paper in any resulting publications, as well as the DOI for the code release: https://zenodo.org/badge/latestdoi/341621577. Scan data are available on our Figshare database: https://figshare.com/projects/Trabecular_and_Cortical_Bone_Segmentation_Method/99434. If you use the scans, please cite this paper as well as the Figshare datasets. The human vertebra dataset [26] is also available on Figshare: https://doi.org/10.6084/m9.figshare.11967897.v1. A video demonstrating trabecular segmentation of a mouse epiphysis is available at: https://doi.org/10.6084/m9.figshare.14137964.v3.

Authors' contributions. E.C.H. designed the study, wrote the manuscript and made the figures. E.C.H., A.A.F. and L.A.E.E. segmented the scans. A.A.F. consulted on the method development. L.A.E.E. and A.A.P. contributed to the study design. S.A. provided scans for the human femoral head and parietal bone. B.J. provided scans for the STR/Ort and CBA mice. E.C.H., A.A.F., L.A.E.E., S.A., B.J. and A.A.P. all edited and revised the manuscript and approve submission of this manuscript.

Competing interests. We declare we have no competing interests.

Funding. We thank the OA Tech+ Network (grant no. EP/N027264/1) and the Anatomical Society (grant no. SSD 011018SEAL-v1-011217) for funding this project. We thank Alessandro Borghi and Great Ormond Street Hospital Charity Clinical Research Starter Grant (grant no. 17DD46) for the human cranial bone scan and Chaozong Liu and the European Commission via H2020-MSCA-RISE programme (BAMOS, grant no. 734156) and Rosetrees Trust (grant no. A1184) for the human femur scan.

Acknowledgements. We thank two anonymous reviewers, whose detailed feedback greatly improved this manuscript. We also thank Kamel Madi for sharing his expertise in Avizo. We thank the Avizo software team, especially Julien Roussel and Alejandra Sanchez-Erostegui, for providing us home office licenses during the COVID pandemic.

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
