## [Peer Review File · Royal Society Open Science]

Review History

RSOS-210408.R0 (Original submission)

Review form: Reviewer 1

Is the manuscript scientifically sound in its present form?

Yes

Are the interpretations and conclusions justified by the results?

Yes

Is the language acceptable?

Yes

Do you have any ethical concerns with this paper?

No

Have you any concerns about statistical analyses in this paper?

No

Recommendation?

Accept with minor revision (please list in comments)

Comments to the Author(s)

See attached pdf (Appendix A).

Review form: Reviewer 2

Is the manuscript scientifically sound in its present form?

Yes

Are the interpretations and conclusions justified by the results?

No

Is the language acceptable?

Yes

Do you have any ethical concerns with this paper?

No

Have you any concerns about statistical analyses in this paper?

No

Recommendation?

Major revision is needed (please make suggestions in comments)

Comments to the Author(s)

SUMMARY

The authors present a method for separating trabecular from cortical bone from high-resolution CT images. The method is tested against different types of samples and the inter-user bias is also analyzed relatively to one part of the method for which manual interaction is required (the so-called sinus cleaning). The method is implemented as a recipe for the commercial software Avizo and the recipe itself is publicly available on GitHub.

GENERAL COMMENTS

The study is well presented and treats a relevant topic, both in terms of the scientific relevance of the task and in terms of the need for increased openness about the methods used. I do not sadly have a license for the software Avizo so I could not test the recipe myself. My comments below are therefore solely based on reading the manuscript and inspecting the datasets on figshare. I hope my comments serve to further improve the manuscript.

The introduction provides a broad panoramic over the existing methodologies for automatic segmentation and separation of cortical and trabecular bone. However, some statements about previous studies do not look particularly clear/accurate. Some examples:

- When talking about the method from Treece and Whitmarsh, which is available through the free tool StradView, it is reported that "...require constant thickness of the cortical bone". I am not sure I understand this, since what StradView does is exactly to identify cortical thickness by probing the intensity profile of the CT image along the normal to the periosteal contour. It can be speculated that StradView requires some settings to be adjusted to the particular case, and that it

can be hard to identify settings that work optimally for the full range of expected cortical thickness. But that is not much different than the recipe presented in this study, in my understanding.

- When citing Väänänen et al. 2019, the authors seem to mix a bit the approach of that study, which essentially uses a single existing periosteal segmentation of the same anatomical compartment (the femur) to automatize the method originally proposed by Treece and Whitmarsh, with the approach by Rueda that instead requires a to fit a statistical shape model over the target image. These two approaches require completely different pre-requisites to run and are based on different techniques.

The authors mention that “In addition to making the Avizo “recipe” freely available, we share its specific steps in Supplementary Information so that the method can also be applied in open-source programs which are more accessible and may be preferred by other researchers.”. That is great, since this helps those readers who do not have an Avizo license. However, I was not able to find such document with all the specific steps. It may very well be that it was provided but I was not able to find it. I could only find one mp4 video and two png images in the supplementary information. Can you please point me to the document with the specific steps?

If I understand it correctly, the preprocessing of the dataset requires manual interaction and separates bone from the background as well as defines the marrow space. That can be seen as a quite relevant part of the whole process, and if that is performed manually it may be hard to call the method completely automated. Some may argue that semi-automated would be more appropriate, in this case.

The manuscript would increase its scientific value if the image sets analyses could be somehow increased. I think it would be great to have the following:

- multiple samples for each anatomical compartment analyzed
- analysis of the performance of the method as a function of different image resolution. If CT scans with different image resolution are not available, it could be worth perhaps downsampling the available images to quantitatively assess the effect of pixel/voxel size on the accuracy of the results.

DETAILED COMMENTS

Page 3, line 30: “cortical: trabecular” should probably be “cortical/trabecular”?

Page 4, section 2.1: I think it would be relevant to provide additional information about the CT images used in this study: manufacturer and model of the CT device used, tube current and kVP just to mention some. I understand reporting all those in the text may make the text hard to read, but perhaps a table would suit the purpose well here?

Page 5, lines 27-32: Do I understand it correctly that the thresholding ranges to separate marrow from bone are to be identified manually? If so, were there any guidelines for the choice of the interval (e.g., based on image histogram)? Additionally, I am wondering what unit of measure the greyscale values have. I assume these values are not calibrated, so they essentially depend on the bit depth of the image and on the scanning parameters used during CT acquisition. If that is the case, I do not see such values as particularly relevant, if not to ensure repeatability on the presented images. They could be removed from main text and placed somewhere in the supplementary material, perhaps?

Page 9, line 14: is the RAM amount the most critical value for execution time here? I suspect CPU and GPU to play a relevant role in this case. If so, it is maybe relevant to also report those specifications.

Page 11, line 46: when claiming “There was no effect”, it may be advisable to add a quantitative metric that supports such a statement.

Page 12, section 3.3: The authors can consider elaborating a bit more on the results of this section. Besides, the statement that inter-user differences are generally low leaves a bit puzzled when 1 out of 3 inter-user analyses reports a 13% difference in trabecular volume / cortical volume. The issue is nicely addressed in the discussion, but the reader is not given any information to elaborate that at this stage of the manuscript.

Page 13, line 16: the authors speculate the method from Ang et al. 2019 would overestimate cortical thickness in 2 of the image sets analyzed in this study. It appears like the code from Ang et al. 2019 is also publicly available, so the authors could consider running that methodology on their image sets to corroborate the speculation.

Page 17, References: it looks like the paper from Väänänen et al, which is referred to in the intro, is not reported in the references. Please check that the reference list fully reflects the papers that have been cited in the main text.

Decision letter (RSOS-210408.R0)

Dear Dr Herbst,

The Editors assigned to your paper RSOS-210408 "A New Straightforward Method for Automated Segmentation of Trabecular Bone from Cortical Bone in Diverse and Challenging Morphologies" have now received comments from reviewers and would like you to revise the paper in accordance with the reviewer comments and any comments from the Editors. Please note this decision does not guarantee eventual acceptance.

Please submit your revised manuscript and required files (see below) no later than 21 days from today's (ie 23-Apr-2021) date. Note: the ScholarOne system will 'lock' if submission of the revision is attempted 21 or more days after the deadline. If you do not think you will be able to meet this deadline please contact the editorial office immediately.

Best regards,
Lianne Parkhouse
Editorial Coordinator
Royal Society Open Science

on behalf of Dr Marco Palanca (Associate Editor) and Kevin Padian (Subject Editor)
openscience@royalsociety.org

Reviewer comments to Author:

Reviewer: 1

Comments to the Author(s)

See attached pdf

Reviewer: 2

Comments to the Author(s)

SUMMARY

The authors present a method for separating trabecular from cortical bone from high-resolution CT images. The method is tested against different types of samples and the inter-user bias is also analyzed relatively to one part of the method for which manual interaction is required (the so-called sinus cleaning). The method is implemented as a recipe for the commercial software Avizo and the recipe itself is publicly available on GitHub.

GENERAL COMMENTS

The study is well presented and treats a relevant topic, both in terms of the scientific relevance of the task and in terms of the need for increased openness about the methods used. I do not sadly have a license for the software Avizo so I could not test the recipe myself. My comments below are therefore solely based on reading the manuscript and inspecting the datasets on figshare. I hope my comments serve to further improve the manuscript.

The introduction provides a broad panoramic over the existing methodologies for automatic segmentation and separation of cortical and trabecular bone. However, some statements about previous studies do not look particularly clear/accurate. Some examples:

- When talking about the method from Treece and Whitmarsh, which is available through the free tool StradView, it is reported that "...require constant thickness of the cortical bone". I am not sure I understand this, since what StradView does is exactly to identify cortical thickness by probing the intensity profile of the CT image along the normal to the periosteal contour. It can be speculated that StradView requires some settings to be adjusted to the particular case, and that it can be hard to identify settings that work optimally for the full range of expected cortical thickness. But that is not much different than the recipe presented in this study, in my understanding.

- When citing Väänänen et al. 2019, the authors seem to mix a bit the approach of that study, which essentially uses a single existing periosteal segmentation of the same anatomical compartment (the femur) to automatize the method originally proposed by Treece and Whitmarsh, with the approach by Rueda that instead requires a to fit a statistical shape model over the target image. These two approaches require completely different pre-requisites to run and are based on different techniques.

The authors mention that "In addition to making the Avizo "recipe" freely available, we share its specific steps in Supplementary Information so that the method can also be applied in open-source programs which are more accessible and may be preferred by other researchers.". That is great, since this helps those readers who do not have an Avizo license. However, I was not able to find such document with all the specific steps. If may very well be that it was provided but I was not able to find it. I could only find one mp4 video and two png images in the supplementary information. Can you please point me to the document with the specific steps?

If I understand it correctly, the preprocessing of the dataset requires manual interaction and separates bone from the background as well as defines the marrow space. That can be seen as a quite relevant part of the whole process, and if that is performed manually it may be hard to call

the method completely automated. Some may argue that semi-automated would be more appropriate, in this case.

The manuscript would increase its scientific value if the image sets analyses could be somehow increased. I think it would be great to have the following:

- multiple samples for each anatomical compartment analyzed
- analysis of the performance of the method as a function of different image resolution. If CT scans with different image resolution are not available, it could be worth perhaps downsampling the available images to quantitatively assess the effect of pixel/voxel size on the accuracy of the results.

DETAILED COMMENTS

Page 3, line 30: “cortical: trabecular” should probably be “cortical/trabecular”?

Page 4, section 2.1: I think it would be relevant to provide additional information about the CT images used in this study: manufacturer and model of the CT device used, tube current and kVP just to mention some. I understand reporting all those in the text may make the text hard to read, but perhaps a table would suit the purpose well here?

Page 5, lines 27-32: Do I understand it correctly that the thresholding ranges to separate marrow from bone are to be identified manually? If so, were there any guidelines for the choice of the interval (e.g., based on image histogram)? Additionally, I am wondering what unit of measure the greyscale values have. I assume these values are not calibrated, so they essentially depend on the bit depth of the image and on the scanning parameters used during CT acquisition. If that is the case, I do not see such values as particularly relevant, if not to ensure repeatability on the presented images. They could be removed from main text and placed somewhere in the supplementary material, perhaps?

Page 9, line 14: is the RAM amount the most critical value for execution time here? I suspect CPU and GPU to play a relevant role in this case. If so, it is maybe relevant to also report those specifications.

Page 11, line 46: when claiming “There was no effect”, it may be advisable to add a quantitative metric that supports such a statement.

Page 12, section 3.3: The authors can consider elaborating a bit more on the results of this section. Besides, the statement that inter-user differences are generally low leaves a bit puzzled when 1 out of 3 inter-user analyses reports a 13% difference in trabecular volume / cortical volume. The issue is nicely addressed in the discussion, but the reader is not given any information to elaborate that at this stage of the manuscript.

Page 13, line 16: the authors speculate the method from Ang et al. 2019 would overestimate cortical thickness in 2 of the image sets analyzed in this study. It appears like the code from Ang et al. 2019 is also publicly available, so the authors could consider running that methodology on their image sets to corroborate the speculation.

Page 17, References: it looks like the paper from Väänänen et al, which is referred to in the intro, is not reported in the references. Please check that the reference list fully reflects the papers that have been cited in the main text.

===PREPARING YOUR MANUSCRIPT===

===PREPARING YOUR REVISION IN SCHOLARONE===

- If you are providing image files for potential cover images, please upload these at this step, and inform the editorial office you have done so. You must hold the copyright to any image provided.
- A copy of your point-by-point response to referees and Editors. This will expedite the preparation of your proof.

- Ensure that your data access statement meets the requirements at <https://royalsociety.org/journals/authors/author-guidelines/#data>. You should ensure that you cite the dataset in your reference list. If you have deposited data etc in the Dryad repository, please include both the 'For publication' link and 'For review' link at this stage.
- If you are requesting an article processing charge waiver, you must select the relevant waiver option (if requesting a discretionary waiver, the form should have been uploaded at Step 3 'File upload' above).
- If you have uploaded ESM files, please ensure you follow the guidance at <https://royalsociety.org/journals/authors/author-guidelines/#supplementary-material> to include a suitable title and informative caption. An example of appropriate titling and captioning may be found at https://figshare.com/articles/Table_S2_from_Is_there_a_trade-off_between_peak_performance_and_performance_breadth_across_temperatures_for_aerobic_scope_in_teleost_fishes_/3843624.

Author's Response to Decision Letter for (RSOS-210408.R0)

See Appendix B.

RSOS-210408.R1 (Revision)

Review form: Reviewer 1

Is the manuscript scientifically sound in its present form?

Yes

Are the interpretations and conclusions justified by the results?

Yes

Is the language acceptable?

Yes

Do you have any ethical concerns with this paper?

No

Have you any concerns about statistical analyses in this paper?

No

Recommendation?

Accept with minor revision (please list in comments)

Comments to the Author(s)

This version of the manuscript has significantly improved, and the concerns of this reviewer have been adequately addressed.

Some residual comments mostly pertain to language and style. In its current form, the manuscript reads as if it was written hastily.

Abstract line 26: "more standardized" - did you mean "reproducible?"

"time-efficient" - did you mean fast/quick?

Line 36: "more porous" is "less dense", so using both is redundant. At the level of bone material however (micrometer scale) trabecular bone has the same density as cortical.

Line 40: please replace "paramount" with a less flamboyant/over-selling adjective.

Line 48-49: It might be better to remove the sentence about "the base of a trabecular column". The flow of the text will be preserved, and the meaning will be clearer. What is a trabecular column anyway? Sounds like an insider term that is specific for the method of acquisition and reconstruction.

Lines 46-58 and other places: Better settle on the hyphenated or non-hyphenated spelling of "time-intensive" ("time intensive"?) and "time-consuming". Same for "shrink-wrapped" later in the text.

Paragraph starting line 71: Stradwin or Stradview? Is it the same thing? The reference doesn't contain it in the title.

Line 175 sealed and non-sealed sounds alright. Repeating "foramen-sealed" is a bit clumsy when repeated multiple times, so it could be explained once at the beginning, and then go with "sealed"?

Line 275: Settle on the tense - past or present, was or is - and keep it consistent throughout the text.

Vaananen reference has a typo. Please check other references spelling and format as well, because it is inconsistent .

Table 1: what is kPV? Did you mean keV or kV?

Downsampling exercise, section 3.4: It is good that the algorithm worked, although the quality of isosurface is low - this is normal, expected, and just confirms once again that about 4 pixels per feature is necessary and sufficient (aka Nyquist rule).

~the end~

Review form: Reviewer 2

Is the manuscript scientifically sound in its present form?

Yes

Are the interpretations and conclusions justified by the results?

Yes

Is the language acceptable?

Yes

Do you have any ethical concerns with this paper?

No

Have you any concerns about statistical analyses in this paper?

No

Recommendation?

Accept with minor revision (please list in comments)

Comments to the Author(s)**SUMMARY**

The authors present a method for separating trabecular from cortical bone from high-resolution CT images. The method is tested against different types of samples and the inter-user bias is also analyzed relatively to one part of the method for which manual interaction is required (the so-called sinus cleaning). The method is implemented as a recipe for the commercial software Avizo and the recipe itself is publicly available on GitHub.

GENERAL COMMENTS

The authors have done a good job at addressing the comments on their original submission. I only have a few minor comments left.

DETAILED COMMENTS

Page 37, line 67-68: "that this is border is"  "that this border is"

Page 37, lines 76-82: The authors sometimes refer to Stradwin, some other times to Stradview. As far as I understand, the two software overlap each other quite a lot in terms of features, but Stradwin is not discontinued and only Stradview is updated. I suggest always referring to Stradview for clarity and consistence. Also, the authors at line 82 write "Stradview method", which I do not think is totally correct. Stradview is the software containing the method, but the method the authors are referring to is most often referred to as cortical bone mapping (CBM), see, e.g., doi: 10.1007/s11914-018-0475-3.

Page 39, line 130: "these "seed""  "these "seeds""

Page 43, line 284: "automatic"  "semi-automatic"

Page 53, table 4: the heading of the table is most likely wrong, I assume "Trabecular BV/TV" should read "Intersection over union", instead.

Decision letter (RSOS-210408.R1)

Dear Dr Herbst

On behalf of the Editors, we are pleased to inform you that your Manuscript RSOS-210408.R1 "A New Straightforward Method for Semi-Automated Segmentation of Trabecular Bone from Cortical Bone in Diverse and Challenging Morphologies" has been accepted for publication in Royal Society Open Science subject to minor revision in accordance with the referees' reports. Please find the referees' comments along with any feedback from the Editors below my signature.

Please submit your revised manuscript and required files (see below) no later than 7 days from today's (ie 25-Jun-2021) date. Note: the ScholarOne system will 'lock' if submission of the revision is attempted 7 or more days after the deadline. If you do not think you will be able to meet this deadline please contact the editorial office immediately.

on behalf of Dr Marco Palanca (Associate Editor) and Kevin Padian (Subject Editor)
openscience@royalsociety.org

Reviewer comments to Author:

Reviewer: 1

Comments to the Author(s)

This version of the manuscript has significantly improved, and the concerns of this reviewer have been adequately addressed.

Some residual comments mostly pertain to language and style. In its current form, the manuscript reads as if it was written hastily.

Abstract line 26: "more standardized" - did you mean "reproducible?"

"time-efficient" - did you mean fast/quick?

Line 36: "more porous" is "less dense", so using both is redundant. At the level of bone material however (micrometer scale) trabecular bone has the same density as cortical.

Line 40: please replace "paramount" with a less flamboyant/over-selling adjective.

Line 48-49: It might be better to remove the sentence about "the base of a trabecular column". The flow of the text will be preserved, and the meaning will be clearer. What is a trabecular column anyway? Sounds like an insider term that is specific for the method of acquisition and reconstruction.

Lines 46-58 and other places: Better settle on the hyphenated or non-hyphenated spelling of "time-intensive" ("time intensive?") and "time-consuming". Same for "shrink-wrapped" later in the text.

Paragraph starting line 71: Stradwin or Stradview? Is it the same thing? The reference doesn't contain it in the title.

Line 175 sealed and non-sealed sounds alright. Repeating "foramen-sealed" is a bit clumsy when repeated multiple times, so it could be explained once at the beginning, and then go with "sealed"?

Line 275: Settle on the tense - past or present, was or is - and keep it consistent throughout the text.

Vaananen reference has a typo. Please check other references spelling and format as well, because it is inconsistent .

Table 1: what is kPV? Did you mean keV or kV?

Downsampling exercise, section 3.4: It is good that the algorithm worked, although the quality of isosurface is low - this is normal, expected, and just confirms once again that about 4 pixels per feature is necessary and sufficient (aka Nyquist rule).
~the end~

Reviewer: 2

Comments to the Author(s)

SUMMARY

The authors present a method for separating trabecular from cortical bone from high-resolution CT images. The method is tested against different types of samples and the inter-user bias is also analyzed relatively to one part of the method for which manual interaction is required (the so-called sinus cleaning). The method is implemented as a recipe for the commercial software Avizo and the recipe itself is publicly available on GitHub.

GENERAL COMMENTS

The authors have done a good job at addressing the comments on their original submission. I only have a few minor comments left.

DETAILED COMMENTS

Page 37, line 67-68: "that this is border is"  "that this border is"

Page 37, lines 76-82: The authors sometimes refer to Stradwin, some other times to Stradview. As far as I understand, the two software overlap each other quite a lot in terms of features, but Stradwin is not discontinued and only Stradview is updated. I suggest always referring to Stradview for clarity and consistence. Also, the authors at line 82 write "Stradview method", which I do not think is totally correct. Stradview is the software containing the method, but the method the authors are referring to is most often referred to as cortical bone mapping (CBM), see, e.g., doi: 10.1007/s11914-018-0475-3.

Page 39, line 130: "these "seed""  "these "seeds""

Page 43, line 284: "automatic"  "semi-automatic"

Page 53, table 4: the heading of the table is most likely wrong, I assume "Trabecular BV/TV" should read "Intersection over union", instead.

===PREPARING YOUR MANUSCRIPT===

===PREPARING YOUR REVISION IN SCHOLARONE===

Author's Response to Decision Letter for (RSOS-210408.R1)

See Appendix C.

Decision letter (RSOS-210408.R2)

Dear Dr Herbst,

I am pleased to inform you that your manuscript entitled "A New Straightforward Method for Semi-Automated Segmentation of Trabecular Bone from Cortical Bone in Diverse and Challenging Morphologies" is now accepted for publication in Royal Society Open Science.

on behalf of Dr Marco Palanca (Associate Editor) and Kevin Padian (Subject Editor)
openscience@royalsociety.org

Appendix A

The manuscript by Herbst and co-authors describes a solution to an interesting problem in bioimaging – the digital separation of cortical and trabecular bone. They describe a digital workflow that indeed sounds robust and useful for a variety of image analysis algorithms. They tested the algorithm on different specimens, namely murine and human bones scans acquired with different voxel sizes. The results appear clean and easy to understand. This topic is timely, but the manuscript could be improved to make it even more interesting for the bioimaging and skeletal biology communities.

Major comment – this submission is extremely verbose for a method paper. The results repeat the intro to a large extent, the discussion repeats the results, and the conclusions present a brief recap of the discussion. The manuscript can be made just twice as short by removing unnecessary repetitions. Major restructuring and distilling is recommended to make this work memorable and citable.

Since this is a method paper, is it possible to make a rich and explicit multipanel figure 1 by adding to the existing figure 1 all the preliminary steps, such as the entire murine tibia with diaphysis, metaphysis, and epiphysis labeled? (and the growth plate, as well.) On a 3D rendered image please illustrate the seeds, and how watershed boundaries coincide with the bone/background boundaries. Then show the metaphysis and the rest, as in the current Fig. 1.

Currently, it is a bit difficult to understand the preprocessing and watershed steps, and also some panels in Fig 1 look mismatching:

Maybe they are accidentally from different slices in the volume. If the authors will improve the first method figure, they could use this opportunity to select exactly the same slices.

In the Discussion, the authors raise an important point – the quality of sampling. Indeed, they used some scans where an average trabecula would be 12 pixels thick, and that is both wonderful and rare. The authors say that their algorithm wouldn't work on a scan where a trabecula is 1-2 pixels thick. This reviewer totally agrees, but indeed not so many analytical algorithms would work on images with poor resolution. Is it possible to run an extra simulation: to downsample one of the high resolution murine scans in increments and repeat the segmentation and check where the accuracy starts deteriorating? Let's say the scan with 5 um pixels can be downsampled x2 several times, to result in the sampling quality of 12, 6, 3 and 1.5 pixels per average trabecular thickness. It would be even more exciting if such downsampling experiment would be done in parallel with the cited Buie method, for more quantitative comparison.

Regarding the accuracy evaluation. The authors use the relative difference % parameter. In fact, there are standard metrics used for reporting segmentation accuracy in a more meaningful way – for example,

comparing unbalanced classes, or different resolutions. Using one of conventional segmentation accuracy metrics will make this work more solid. Here is a useful reference that describes various mainstream metrics <https://doi.org/10.1186/s12880-015-0068-x>

Maybe intersection-over-union, or the Dice index would be appropriate for this submitted manuscript?

All comments below are minor/technical

This reviewer encourages the authors to double-check the anatomical terminology. Is it surely “sinuses”, and not nutritive foramina (singular – foramen)? Or maybe these are “openings” or “inlets” (imagine if one of the readers will try to segment a bird humerus with a large airsac cavity...)

As well, “sinus cleaning” sounds like a medical procedure. Did you mean opening/foramen/inlet obturation? Or sealing? Or obstruction?

Section 2.2.1, preprocessing, end of paragraph 2. Are the reported values the grayscale values of 8-bit images? Can this be corroborated by lower Otzu segmentation? What about 16 bit images?

Can you specify the purpose of filtering? Is that because the images were noisy? What were the scanning parameters? Instrument, voltage etc.

Is it possible to give a clear definition of shrink-wrapping? Maybe with a self-explanatory diagram?

The smoothing kernel (aka ball) size – is it related to the resolution of the scans? Would it be the same in high-resolution and medium-resolution images. Was it the most conservative smallest kernel regardless of the image sampling? Please add a one-sentence rationale for the kernel size selection.

The authors call the 3D representations of trabeculae in Fig 3 B, D, F and Fig 4 B and D – a “model”. Is it not a surface rendering?

Of positive remarks – this reviewer is very much impressed by the speed of the described algorithm.

Appendix B

We thank the reviewers and the editors for considering our manuscript, pending major revisions. We greatly appreciate the thorough comments of both reviewers, which helped improve the manuscript. We have compiled the reviewer comments in this sheet and added our responses in blue text. For ease of viewing, figures are still included in-text in the “tracked changes” version of the revised manuscript, but are uploaded separately from the “changes accepted” document as required by RSOS. Line numbers in responses below refer to the “tracked changes” version with “all markup” selected.

>>>>

Reviewer comments to Author:

Reviewer: 1. Comments to the Author(s)

The manuscript by Herbst and co-authors describes a solution to an interesting problem in bioimaging – the digital separation of cortical and trabecular bone. They describe a digital workflow that indeed sounds robust and useful for a variety of image analysis algorithms. They tested the algorithm on different specimens, namely murine and human bones scans acquired with different voxel sizes. The results appear clean and easy to understand. This topic is timely, but the manuscript could be improved to make it even more interesting for the bioimaging and skeletal biology communities.

Thank you for your feedback and useful comments that greatly improved the manuscript.

1. Major comment – this submission is extremely verbose for a method paper. The results repeat the intro to a large extent, the discussion repeats the results, and the conclusions present a brief recap of the discussion. The manuscript can be made just twice as short by removing unnecessary repetitions. Major restructuring and distilling is recommended to make this work memorable and citable.

Our intention was to ensure that the method could be adopted by the expert and the non-expert imager alike, and that the repetition is important to ensure that the non-expert readily understands the method and its rationale. For example, reviewer 11 wanted *more* elaboration in the Results by mentioning one of the Discussion points in the Results section already (see point 11) However, we do agree that the paper is quite lengthy and therefore made various edits to make our explanations more concise (most of these are visible in red marked changes). We also removed some sections, for example part of the Intro (lines 99-102) and the Conclusion (lines 584-589). We also restructured the Discussion to make the points (especially discussing the Buie et al. method) more linear and concise (459-515). To aid in readability and reduce the length of the text, we also moved the scan thresholding, and data availability information to Table 1 (section 2.1)

2. Since this is a method paper, is it possible to make a rich and explicit multipanel figure 1 by adding to the existing figure 1 all the preliminary steps, such as the entire murine tibia with diaphysis, metaphysis, and epiphysis labeled? (and the growth plate, as well.) On a 3D rendered image please illustrate the seeds, and how watershed boundaries coincide with the bone/background boundaries. Then show the metaphysis and the rest, as in the current Fig. 1.

We expanded Figure 1 to include panels showing the proximal tibia including epiphysis, growth plate, and diaphysis (Fig. 1A), an example of seed placement (Fig. 1B), a 2D image

of the resulting boundaries of the watershed operation (Fig. 1C), and the marrow space isolated via thresholding after the watershed operation (Fig. 1D). We also added references to these extra panels in the text. Please note we changed the specimen from a 19 week STR/Ort mouse to a 20 week CBA mouse for this figure because we had not saved the seed layer in the processing of data from the STR/Ort mouse. We included 2D examples of seeds in 2 planes rather than a 3D rendering because in the 3D rendering it is not easy to see the relationship between the seeds and the anatomy, especially because we drew very big seeds in the exterior (because it is easy to select large areas, for example slices with just background; this is not any more time-consuming than placing small seeds). We also made a note about seed placement in lines 213-215.

3. Currently, it is a bit difficult to understand the preprocessing and watershed steps, and also some panels in Fig 1 look mismatching:

Maybe they are accidentally from different slices in the volume. If the authors will improve the first method figure, they could use this opportunity to select exactly the same slices.

Thank you for noticing this. We agree that it makes sense to use consistent slices. We have updated Figure 1 accordingly. We have also made more explicit reference to the Figure 1 in order to make the preprocessing and watershed steps easier to understand (sections 2.2.1 and 2.2.2).

4. In the Discussion, the authors raise an important point – the quality of sampling. Indeed, they used some scans where an average trabecula would be 12 pixels thick, and that is both wonderful and rare. The authors say that their algorithm wouldn't work on a scan where a trabecula is 1-2 pixels thick. This reviewer totally agrees, but indeed not so many analytical algorithms would work on images with poor resolution. Is it possible to run an extra simulation: to downsample one of the high resolution murine scans in increments and repeat the segmentation and check where the accuracy starts deteriorating? Let's say the scan with 5 um pixels can be downsampled x2 several times, to result in the sampling quality of 12, 6, 3 and 1.5 pixels per average trabecular thickness. It would be even more exciting if such downsampling experiment would be done in parallel with the cited Buie method, for more quantitative comparison.

We really appreciate this excellent idea to take the same scans and downsample them to test the algorithm on lower resolution scans. We took one of our mouse tibiae datasets and downsampled it several times (lines 293-305 and Table 2). The results are reported in lines 401-413 and Figure 6 and discussed in lines 603-610. We scaled down the algorithm proportionally to the downsampling and generally got good results with this approach, even on the 1.66 pixel trabeculae (although they all, as expected, deviated slightly from the higher resolution scan). We agree that also running this experiment on the Buie et al. method would

be informative but unfortunately we did not have the time. We have nonetheless made the point to emphasise this opportunity in the modified version (lines 608-610).

5. Regarding the accuracy evaluation. The authors use the relative difference % parameter. In fact, there are standard metrics used for reporting segmentation accuracy in a more meaningful way – for example, comparing unbalanced classes, or different resolutions. Using one of conventional segmentation accuracy metrics will make this work more solid. Here is a useful reference that describes various mainstream metrics <https://doi.org/10.1186/s12880-015-0068-x> Maybe intersection-over-union, or the Dice index would be appropriate for this submitted manuscript?

We thank the reviewer for this excellent suggestion and the useful paper discussing these metrics. We wrote an ImageJ macro to implement the intersection over union approach (lines 259-262). Results are shown in Table 4 and lines 342-343, 396-397 and we share the code for this ImageJ macro (line 262).

All comments below are minor/technical

6. This reviewer encourages the authors to double-check the anatomical terminology. Is it surely “sinuses”, and not nutritive foramina (singular – foramen)? Or maybe these are “openings” or “inlets” (imagine if one of the readers will try to segment a bird humerus with a large airsac cavity...)

We agree with the reviewer that the word “sinus” is not optimal. Therefore, we have changed this term to “foramen/foramina” throughout the manuscript.

7. As well, “sinus cleaning” sounds like a medical procedure. Did you mean opening/foramen/inlet obturation? Or sealing? Or obstruction?

We agree with the viewer and therefore changed “cleaning” to “sealing”, and explained that “sealing” means removing the sinuses from the marrow space segmentation (line 227)

8. Section 2.2.1, preprocessing, end of paragraph 2. Are the reported values the grayscale values of 8-bit images? Can this be corroborated by lower Otsu segmentation? What about 16 bit images?

The images are 8 bit images and the reported thresholds were greyscale values for 8 bit images. We added this information to Table 1 (and moved the greyscale thresholds alongside instead of reporting them in the text). We were not aware of the Otsu method at the time of writing the paper but checked and this is possible in Avizo. In line with the reviewer’s recommendation we sought to corroborate our data for one of our samples (20 week CBA tibia) and the Otsu segmentation found that the marrow space was 0-80 (out of 255 greyscale values) whereas our threshold range was 0-70. For the marrow segmentation that was a volume difference of 3.13%. Future studies could use this automatic method to segment rather than use our manual selection. However, for the sake of our algorithm as long as a reasonable threshold is chosen that separates the marrow space from the bone, it does not matter whether we use the Otsu method or not (as long as the same threshold is used for all samples and for all users). Therefore we did not report the outcome of the Otsu method in our paper, because we just tested this retroactively. We thank the reviewer for this suggestion because it is a very useful tool

that we expect to use in the future. We also added 2 sentences in section 4.1 recommending that the readers make use of this helpful tool (lines 621-624).

9. Can you specify the purpose of filtering? Is that because the images were noisy?

This filtering was indeed done to remove noise - filtering helps with the watershed operation, making the boundaries “smoother”. We added this info into section 2.2.1 line 157.

10. What were the scanning parameters? Instrument, voltage etc.
Thank you for the suggestion. We added a table of scanning parameters (new Table 1).

11. Is it possible to give a clear definition of shrink-wrapping? Maybe with a self-explanatory diagram?

Shrink wrapping is shown in Figure 1F. Our text also included a definition of the shrink wrapping in terms of the operations used, but to clarify we have provided an additional summary of this step (lines 190-191).

12. The smoothing kernel (aka ball) size – is it related to the resolution of the scans? Would it be the same in high-resolution and medium-resolution images. Was it the most conservative smallest kernel regardless of the image sampling? Please add a one-sentence rationale for the kernel size selection.

In section 2.2.2 we noted that the smoothing kernel was determined via testing the algorithm during development: *“This value was chosen based on iterative tests on the mouse samples, and worked well for the human samples (which had different resolutions)”* (lines 192-193).

13. The authors call the 3D representations of trabeculae in Fig 3 B, D, F and Fig 4 B and D – a “model”. Is it not a surface rendering? Of positive remarks – this reviewer is very much impressed by the speed of the described algorithm.

It is indeed a volume rendering and so we have updated the captions accordingly.

Reviewer: 2. Comments to the Author(s)

SUMMARY

The authors present a method for separating trabecular from cortical bone from high-resolution CT images. The method is tested against different types of samples and the inter-user bias is also analyzed relatively to one part of the method for which manual interaction is required (the so-called sinus cleaning). The method is implemented as a recipe for the commercial software Avizo and the recipe itself is publicly available on GitHub.

GENERAL COMMENTS

1. The study is well presented and treats a relevant topic, both in terms of the scientific relevance of the task and in terms of the need for increased openness about the methods used. I do not sadly have a license for the software Avizo so I could not test the recipe myself. My comments below are therefore solely based on reading the manuscript and inspecting the datasets on figshare. I hope my comments serve to further improve the manuscript.

Thank you, your comments greatly improve the manuscript. We are especially thankful for catching the errors in our interpretations/descriptions of other methods.

2. The introduction provides a broad panoramic over the existing methodologies for automatic segmentation and separation of cortical and trabecular bone. However, some statements about previous studies do not look particularly clear/accurate. Some examples:

2.1. When talking about the method from Treece and Whitmarsh, which is available through the free tool StradView, it is reported that "...require constant thickness of the cortical bone". I am not sure I understand this, since what StradView does is exactly to identify cortical thickness by probing the intensity profile of the CT image along the normal to the periosteal contour. It can be speculated that StradView requires some settings to be adjusted to the particular case, and that it can be hard to identify settings that work optimally for the full range of expected cortical thickness. But that is not much different than the recipe presented in this study, in my understanding.

Thank you for noticing this error. Whitmarsh 2019 did indeed erode the cortex in the femoral head by a constant value to remove the cortex and isolate the trabeculae (see "Global Evaluation" section in their paper) but you are correct that we mistakenly misrepresented the StradView method developed by Treece et al. 2010. We realize that this method was a separate component of the Whitmarsh 2019 paper and, as you pointed out, that the StradView method is used to detect thickness changes in cortical bone. We have corrected our mistake and removed the Treece et al. 2010 reference here (line 70), and moved the discussion of the Treece method to the section discussing the Väänänen et al. 2019 method (see response to point 2.2).

2.2. When citing Väänänen et al. 2019, the authors seem to mix a bit the approach of that study, which essentially uses a single existing periosteal segmentation of the same anatomical compartment (the femur) to automatize the method originally proposed by Treece and Whitmarsh, with the approach by Rueda that instead requires a to fit a statistical shape model over the target image. These two approaches require completely different pre-requisites to run and are based on different techniques.

Thank you very much for catching this error and for emphasizing the difference between the approaches. We rewrote this section with more detail on both methods (lines 75-94), and noted that while Väänänen et al. 2019 used a template to automate the Treece method, the Treece method is also possible without this template, as long as the periosteal surface is determined otherwise (for example with contours). We misinterpreted the automated part of the method in Väänänen et al. 2019 and realize now that this reference was used to automate the detection of the periosteal surface, but that the StradView algorithm itself does not require this. We hope that our changes now emphasise that the two approaches require completely different pre-requisites to run and are based on different techniques.

3. The authors mention that "In addition to making the Avizo "recipe" freely available, we share its specific steps in Supplementary Information so that the method can also be applied in open-source programs which are more accessible and may be preferred by other researchers.". That is great, since this helps those readers who do not have an Avizo license. However, I was not able to find such document with all the specific steps. It may very well be that it was provided but I was not able to find it. I could only find one mp4 video and two png

images in the supplementary information. Can you please point me to the document with the specific steps?

Thank you very much for noticing this error. We initially planned to include this info in the Supplementary Information but then migrated everything to Github to make it easier to access (https://github.com/evaherbst/Trabecular_Segmentation_Avizo). The Github link was already present in section 6 but we forgot to update the section, as the reviewer recognised. We have now replaced “Supplementary Information” with the Github repository link (line 440). We also created a code release and added a DOI for the most recent code release in lines 198 and 652 and added the DOI for the human vertebra dataset in section 6, lines 654-655.

4. If I understand it correctly, the preprocessing of the dataset requires manual interaction and separates bone from the background as well as defines the marrow space. That can be seen as a quite relevant part of the whole process, and if that is performed manually it may be hard to call the method completely automated. Some may argue that semi-automated would be more appropriate, in this case.

We agree, and have adjusted the title and mentions of “automated” in the text accordingly. We also added in a clarification about the preprocessing versus automated parts in the Introduction (lines 108-111).

5. The manuscript would increase its scientific value if the image sets analyses could be somehow increased. I think it would be great to have the following:

- multiple samples for each anatomical compartment analyzed

Our sample size is similar to that use in Ang et al. 2019, although we acknowledge that Buie et al. 2007 and Burghardt 2010 had much larger dataset. We are currently analyzing multiple other murine samples for another study and are finding that the algorithm is working well; these will be published in a future paper (Herbst, Evans et al. in prep). We did not have time to include further human samples, but we believe that with our choice of specimens of varying species, ages, and pathologies, our method is sufficiently validated. We have nonetheless added a comment to the discussion in recognition of the point that an even greater range of sample anatomies/types will only increase the study’s scientific value further (lines 430-431).

- analysis of the performance of the method as a function of different image resolution. If CT scans with different image resolution are not available, it could be worth perhaps downsampling the available images to quantitatively assess the effect of pixel/voxel size on the accuracy of the results.

We agree with this idea and therefore tested the algorithm on lower resolution samples. Please see our response to Reviewer 1’s comment #4.

DETAILED COMMENTS

6. Page 3, line 30: “cortical: trabecular” should probably be “cortical/trabecular”?

We clarified the text, replacing it with “interface between the cortical and trabecular bone” (line 64) for clarification, since cortical/trabecular could be misinterpreted as “cortical or trabecular” or “cortical divided by trabecular”.

7. Page 4, section 2.1: I think it would be relevant to provide additional information about the CT images used in this study: manufacturer and model of the CT device used, tube current and kVP just to mention some. I understand reporting all those in the text may make the text hard to read, but perhaps a table would suit the purpose well here?

Thank you for the suggestion. We added a table of scanning parameters (new Table 1).

8. Page 5, lines 27-32: Do I understand it correctly that the thresholding ranges to separate marrow from bone are to be identified manually? If so, were there any guidelines for the choice of the interval (e.g., based on image histogram)? Additionally, I am wondering what unit of measure the greyscale values have. I assume these values are not calibrated, so they essentially depend on the bit depth of the image and on the scanning parameters used during CT acquisition. If that is the case, I do not see such values as particularly relevant, if not to ensure repeatability on the presented images. They could be removed from main text and placed somewhere in the supplementary material, perhaps?

Yes, you have understood this correctly. We chose the intervals based on what appeared to be the best segmentation to us (without the use of image histograms) – we clarified that these thresholds were chosen manually in line 175. Regarding the images, we added in information that all images were 8 bit (lines 136-137) and added the table of scanning parameters (Table 1). With this information we think the threshold parameters are still useful to include here if anyone wants to recreate the segmentation on these specific images (which we shared on Figshare). We also moved the greyscale value information to the same table. Reviewer 1 also asked about the thresholding and recommended using the Otsu automatic thresholding method – please see our response to point 8 from Reviewer 1 for more information on this thresholding comparison.

9. Page 9, line 14: is the RAM amount the most critical value for execution time here? I suspect CPU and GPU to play a relevant role in this case. If so, it is maybe relevant to also report those specifications.

We added the CPU and GPU specifications (line 319).

10. Page 11, line 46: when claiming “There was no effect”, it may be advisable to add a quantitative metric that supports such a statement.

Yes, we agree. The difference was in trabecular volume was 0% and we added this information to the manuscript (line 372), as well as clarifying that Figure 5F shows trabeculae produced by both inputs (one including these horizontal foramina and one excluding them) in line 386.

11. Page 12, section 3.3: The authors can consider elaborating a bit more on the results of this section. Besides, the statement that inter-user differences are generally low leaves a bit puzzled when 1 out of 3 inter-user analyses reports a 13% difference in trabecular volume / cortical volume. The issue is nicely addressed in the discussion, but the reader is not given any information to elaborate that at this stage of the manuscript.

We agree that this point should already be mentioned in the results and that more detail is needed; we added more information to section 3.3 accordingly (lines 391-397).

12. Page 13, line 16: the authors speculate the method from Ang et al. 2019 would overestimate cortical thickness in 2 of the image sets analyzed in this study. It appears like the code from Ang et al. 2019 is also publicly available, so the authors could consider running that methodology on their image sets to corroborate the speculation.

We agree with the reviewer that this would be a good test but unfortunately given time constraints we are not able to test this ourselves. We added further details on why we think that we would get these type of errors, but also added a note that further tests are needed to support our speculation (lines 449-455).

13. Page 17, References: it looks like the paper from Väänänen et al, which is referred to in the intro, is not reported in the references. Please check that the reference list fully reflects the papers that have been cited in the main text.

Thank you for noticing this error. We added the Väänänen et al. paper to the references (lines 766-768).

Appendix C

We thank the reviewers and the editors for accepting the manuscript with minor revisions. We thank the reviewers for their feedback to further improve the text. We have compiled the minor reviewer comments in this sheet and added our responses in blue text.

Reviewer: 1

Comments to the Author(s)

1. Abstract line 26: "more standardized" - did you mean "reproducible"?
Yes, we changed this to "reproducible"
2. "time-efficient" - did you mean fast/quick?
Yes, we specified fast, and specified (relative to manual segmentation) when discussing this speed.
3. Line 36: "more porous" is "less dense", so using both is redundant. At the level of bone material however (micrometer scale) trabecular bone has the same density as cortical.
We agree with both of these points and changed this section to "is more porous at the tissue level"
4. Line 40: please replace "paramount" with a less flamboyant/over-selling adjective.
We took out the adjective and instead just wrote "Due to these differences, many studies investigate the..."
5. Line 48-49: It might be better to remove the sentence about "the base of a trabecular column". The flow of the text will be preserved, and the meaning will be clearer. What is a trabecular column anyway? Sounds like an insider term that is specific for the method of acquisition and reconstruction.
We mean the region where the trabecula meets the cortical bone – we left out this sentence and also slightly changed the previous sentence for clarity to: "In particular, it is difficult to determine where the cortical bone ends and the trabecular bone begins."
6. Lines 46-58 and other places: Better settle on the hyphenated or non-hyphenated spelling of "time-intensive" ("time intensive"?) and "time-consuming". Same for "shrink-wrapped" later in the text.
We changed these to exclude the hyphens.
7. Paragraph starting line 71: Stradwin or Stradview? Is it the same thing? The reference doesn't contain it in the title.
We clarified this - see response to Reviewer 2 point 2

8. Line 175 sealed and non-sealed sounds alright. Repeating "foramen-sealed" is a bit clumsy when repeated multiple times, so it could be explained once at the beginning, and then go with "sealed"?
Good idea, we made this change.
9. Line 275: Settle on the tense - past or present, was or is - and keep it consistent throughout the text.
We changed "is" to "was" – also changed to past tense in lines 79, 136, 137, 235, 242. However, we kept the present tense when discussing details of our algorithm in the methods as well as when discussing our methods and other methods in the Discussion, since these methods still exist (so "are" and "is" seems more correct than "were" and "was").
10. Vaananen reference has a typo. Please check other references spelling and format as well, because it is inconsistent .
We corrected the typo and the format of this reference, and added DOIs to those references that were missing a DOI.
11. Table 1: what is kPV? Did you mean keV or kV?
Thank you for catching this typo. We mean kV and changed it accordingly

Reviewer: 2

Comments to the Author(s)

1. Page 37, line 67-68: "that this is border is"  "that this border is"
We deleted the first "is".
2. Page 37, lines 76-82: The authors sometimes refer to Stradwin, some other times to Stradview. As far as I understand, the two software overlap each other quite a lot in terms of features, but Stradwin is not discontinued and only Stradview is updated. I suggest always referring to Stradview for clarity and consistence. Also, the authors at line 82 write "Stradview method", which I do not think is totally correct. Stradview is the software containing the method, but the method the authors are referring to is most often referred to as cortical bone mapping (CBM), see, e.g., doi: 10.1007/s11914-018-0475-3.

You are right that Stradwin was the original software that is no longer maintained – thank you for making us aware of this. We decided to refer to Stradwin in the text (since that is the software those papers used), but we added a note in the text to tell readers that Stradwin is now replaced by Stradview. We also changed "Stradview method" (we meant the segmentation method using the Stradview software, but realize that our wording incorrect). Lines 82-73 now state "The Stradwin software can also be used without a template to segment bone"
3. Page 39, line 130: "these "seed""  "these "seeds""
We changed "seed" to "seeds".
4. Page 43, line 284: "automatic"  "semi-automatic"

We changed “automatic” to “semi-automatic”

5. Page 53, table 4: the heading of the table is most likely wrong, I assume “Trabecular BV/TV” should read “Intersection over union”, instead.

Yes, we changed “trabecular BV/TV” to “Intersection over Union”